# Kinetic Energy-Based Indicators to Compare Different Load Models of a Mobile Crane

**DOI:** 10.3390/ma15228156

**Published:** 2022-11-17

**Authors:** Andrzej Urbaś, Krzysztof Augustynek, Jacek Stadnicki

**Affiliations:** Department of Mechanical Engineering Fundamentals, Faculty of Mechanical Engineering and Computer Science, University of Bielsko-Biala, Willowa 2, 43-306 Bielsko-Biala, Poland

**Keywords:** mobile crane, dynamics, joint coordinates, load modeling method, energy indicators

## Abstract

This paper presents a proposal for the use of energy indicators to evaluate the load modelling methods on the dynamics of a mobile crane. Three different variants of mathematical models of a load carried were examined and compared: as a lumped mass on one hook-sling, as a sphere on one hook-sling, and as a box on four hook-slings. The formalism of joint coordinates and homogeneous transformation matrices were applied to define the kinematics of the system. The equations of motion were derived using the Lagrange equations of the second kind. These equations were supplemented by the Lagrange multipliers and constraint equations formulated for the cut-joints and drives. The energy indicators were proposed to evaluate the behavior of the crane and the carried load. The authors proved that modeling a load in the form of a lumped mass is a great simplification in the analysis of crane dynamics.

## 1. Introduction

In literature, there are many works devoted to the application of mathematical models in which the movement of a load is analyzed. One can find a review of the mathematical models and control strategies of cranes in [1]. The authors classify them and discuss their applications and limitations. Depending on the application, these models are more or less extensive. Usually, simplified models—with regard to the crane structure and the load—are used for control and optimization issues [2,3]. The load is then modeled as a lumped mass suspended on a rigid or flexible rope [4].

Contrary to the large number of papers on the mathematical models of cranes in which the load is modeled as a lumped mass (with three dofs), there are few studies in which the load is treated as a rigid body with six dofs. The theory of rigid body dynamics is applied when the load has six dofs. Some papers deal with special cases of rigid body motion; the spatial motion of a symmetric rigid body about a fixed point subjected to perturbing torques, a gyro moment vector, and Newtonian field forces are analyzed in [5]. Similar considerations were shown in [6], where authors slightly shifted the center mass of a rigid body with respect to the principal axis of dynamic symmetry.

Rigid body load models are used when the dimensions of the load affect the dynamics, stability, or steering of the crane. In addition, the crane dynamics models may take into account other phenomena affecting its operation, such as the motion of the base, the flexibility of the load suspension system (rope(s) system, grab), wind action, etc. The dynamics analysis of a load suspended by a floating crane was presented in [7]. The authors defined the equations of the motions of the floating crane (six dofs) and the load (six dofs), which contained the coupled equations resulting from the motion of the floating crane and the load connected by rope. The proposed model was applied to estimate the motion of the floating crane and the load and also to calculate the tension of the rope. A multibody system that consisted of a floating crane barge, an elastic boom, and a load connected to the boom through ropes was presented in [8]. In the presented model, the floating crane, as well as the load, had six dofs, and the elastic boom deformed in three dimensions. The influence of the boom flexibility on the load lifting system was analyzed and compared. A method for the determination of the dynamic reaction forces in knuckle boom cranes was proposed in [9]. The method was based on the dynamic modeling procedure, which is an extension of Kane’s equations of motion. The same authors in [10] formulated a procedure for modeling a flexible knuckle boom crane, which was actuated by means of the hydraulic cylinders. The presented model was a planar multibody system. A method of calculating the reaction forces of a crane installed on a ship that could move in the conditions of sea waves was presented in [11]. The method was used to calculate the reaction forces between the crane base and the vessel deck. The proposed method included a case where the crane was mounted on a platform that kept the base of the crane horizontal when the vessel was rolling and pitching. In [12], the authors presented a dynamics model of bridge cranes transporting heterogeneous loads while taking into account the eccentricity between the mass center and centroid of the load. The authors compared the simulation and experimental results obtained from a small-scale bridge crane transporting a heterogeneous bar. A stability assessment method of the mobile crane handling system based on the safety indicator was presented in [13]. The proposed mathematical model was built in the integrated CAD/CAE environment. The model allows the analysis of the displacements of the mass center of the crane system, the reactions of the outrigger system, and the stabilizing and overturning torques that act on the crane, as well as the safety indicator values for the given movement trajectories of the crane’s working elements. The same authors in [14] used the same crane model together with computational intelligence methods for the analysis and simulation of a crane system during sequential movements. Neural-network-supported analysis of varying contact forces exerted by the outriggers onto the ground showed that the contact forces stabilize and overturn torques, and the mass center during the handling was allowed to specify trajectories, ensuring the stability of the crane. In [15] the authors proposed the dynamics model of spatial container cranes subjected to wind loads. The container was modeled as a rigid body, elastically suspended from the trolley and running along rails on top of the crane boom and the girder. The analyzed system was identified and validated with full-scale experimental tests. The analysis of the load motion, after taking into account wind pressure and the deformation of the rope system of the mobile crane, is presented in [16]. The carried load was treated as a rigid body. The Kelvin–Voigt model was applied to model the rope system. The same authors in [17,18] proposed a method for determining the effective surface area of a rigid body induced by the wind. The Cardan angles and its derivatives were used to determine the rigid body orientation in space.

In the case of grab cranes, the carried load is considered to be part of the grab, and its dimensions have great importance on dynamics. The dynamics model of the forest crane and the lifted load modeled as a 3D rigid body is presented in [19]. The model enabled the analysis of the motion of the load carried by a forest crane while taking into account the elastic deformations of the boom. The mathematical model of the forest crane applied to output tracking motion control for a forestry crane actuated by hydraulic cylinders was proposed in [20]. The proposed control system presented challenges related to flexibility, oscillatory response, and unpredictable model uncertainty. The dynamics model of the grab crane that took into account the friction in the joint was presented in [21]. Numerical analyses performed presented the influences of various friction forces on the vibration level, as perceived by the operator of the crane. The level of discomfort was discussed based on standards commonly used in the vehicle and transportation industry for evaluations of vibration comfort.

The main contribution of this work is to indicate that the use of a mathematical model of a crane with a load in the form of a lumped mass may not reflect the real dynamics of the crane. The energy indicators are proposed to evaluate the behavior of the crane and the carried load.

The selected results presented in this paper were presented at the 30th Conference Vibrations in Physical Systems—VIBSYS 2022, 26–28 September 2022, Poznań, Poland [22].

## 2. Mathematical Models of the System

The considered mobile crane was divided into four subsystems (Figure 1):Crane suspension subsystem. This system was modeled in the form of the flexible, supported base mounted on the ground by means of the system of wheels and outriggers. The wheels and outriggers systems were modeled using spring-damping elements.Supporting structure. This system contained the rotary column, two boom systems, and the telescopic boom section. It was modeled in the form of the open-loop kinematic chain (called the main chain—
$c_{m}$).Main load lifting subsystems. These systems refer to the hydraulic cylinders. They were modeled in the form of the closed-loop kinematic chain (called the auxiliary chains—
${c_{a,\alpha }|}_{\alpha =1,2}$).Load suspension subsystem. A carried load was modeled in three variants: as a lumped mass on one hook-sling (3 dofs), as a sphere on one hook-sling (6 dofs), and as a box on four hook-slings (6 dofs). The rope was modeled using spring-damping elements.

The selected links of the crane were driven using kinematic input.

The assumed values of geometrical parameters are shown in Table 1.

Joint coordinates and homogeneous transformation matrices [23,24] were applied to describe the kinematics of the crane considered, as shown in Figure 2.

The generalized coordinates vector is defined as follows
(1)
$$q=\left[\begin{array}{ccc} q^{{\left(c\right)}^{T}} & ¦ & q^{{\left(l\right)}^{T}} \end{array}\right]={\left[\begin{array}{cccccc} q^{{\left(b\right)}^{T}} & q^{{\left(c_{m}\right)}^{T}} & {\bar{q}}^{{\left(c_{a,1}\right)}^{T}} & {\bar{q}}^{{\left(c_{a,2}\right)}^{T}} & ¦ & q^{{\left(l\right)}^{T}} \end{array}\right]}^{T},$$
where:
$q^{\left(c\right)},\;\;q^{\left(l\right)}$ are the vectors defining the motion of the crane and the load,
$q^{\left(b\right)}$ is the vector describing the motion of the base, 
$q^{\left(b\right)}={\left[\begin{array}{cccccc} {\hat{x}}^{\left(b\right)} & {\hat{y}}^{\left(b\right)} & {\hat{z}}^{\left(b\right)} & {\hat{\psi }}^{\left(b\right)} & {\hat{\theta }}^{\left(b\right)} & {\hat{\varphi }}^{\left(b\right)} \end{array}\right]}^{T}$,
$q^{\left(c_{m}\right)}$ is the vector defining the motion of the supporting structure of the crane,
$q^{\left(c_{m}\right)}={\left[\begin{array}{cccc} {\hat{\psi }}^{\left(c_{m},l_{1}\right)} & {\hat{\psi }}^{\left(c_{m},l_{2}\right)} & {\hat{\psi }}^{\left(c_{m},l_{3}\right)} & {\hat{z}}^{\left(c_{m},l_{4}\right)} \end{array}\right]}^{T}$,and 
${\bar{q}}^{\left(c_{a,1}\right)},\;\;{\bar{q}}^{\left(c_{a,2}\right)}$ are the vectors describing the motions of the main load lifting systems (hydraulic cylinders). 
${\bar{q}}^{\left(c_{a,1}\right)}={\left[\begin{array}{cc} {\hat{\psi }}^{\left(c_{a,1},l_{1}\right)} & {\hat{z}}^{\left(c_{a,1},l_{2}\right)} \end{array}\right]}^{T}$, 
${\bar{q}}^{\left(c_{a,2}\right)}={\left[\begin{array}{cc} {\hat{\psi }}^{\left(c_{a,2},l_{1}\right)} & {\hat{z}}^{\left(c_{a,2},l_{2}\right)} \end{array}\right]}^{T}$, and
$q^{\left(l\right)}=\{\begin{array}{ccc} \;\;\;\;\;\;\;\;\;\;\;{\left[\begin{array}{ccc} x^{\left(l_{m}\right)} & y^{\left(l_{m}\right)} & z^{\left(l_{m}\right)} \end{array}\right]}^{T} & \mathrm{if} & \mathrm{lumped}\;\;\mathrm{mass},\\ \;{\left[\begin{array}{cccccc} x^{\left(\alpha \right)} & y^{\left(\alpha \right)} & z^{\left(\alpha \right)} & \psi^{\left(\alpha \right)} & \theta^{\left(\alpha \right)} & \varphi^{\left(\alpha \right)} \end{array}\right]}_{\alpha \in \left\{l_{s},l_{b}\right\}}^{T} & \mathrm{if} & \mathrm{sphere}\;\mathrm{or}\;\mathrm{box}. \end{array}$

The dynamics equations of motion were derived using the Lagrange equations of the second kind [23], and they can be written in the following matrix form
(2)
$$\left[\begin{array}{cccc} M^{\left(c\right)} & 0 & C^{{\left(j\right)}^{T}} & C^{{\left(d\right)}^{T}}\\ 0 & M^{\left(l\right)} & 0 & 0\\ C^{\left(j\right)} & 0 & 0 & 0\\ C^{\left(d\right)} & 0 & 0 & 0 \end{array}\right]\left[\begin{array}{c} {\ddot{q}}^{\left(c\right)}\\ {\ddot{q}}^{\left(l\right)}\\ f^{\left(j\right)}\\ f^{\left(d\right)} \end{array}\right]=\left[\begin{array}{c} f^{\left(c\right)}\\ f^{\left(l\right)}\\ d^{\left(j\right)}\\ d^{\left(d\right)} \end{array}\right],$$
where

$M^{\left(c\right)},\;\;M^{\left(l\right)}$ are the mass matrices of the crane and load,
$C^{\left(j\right)},\;\;C^{\left(d\right)}$ are the constraint matrices related to the cut-joints and drives,
$f^{\left(j\right)}$ is the vector of unknown reaction forces in the cut-joints,
$f^{\left(d\right)}$ is the vector of the unknown driving torque and forces,
$f^{\left(c\right)},\;\;f^{\left(l\right)}$ are the vectors of the right side of the dynamics equations,
$f^{\left(c\right)}=-e^{\left(c\right)}-g^{\left(c\right)}-s^{\left(s\right)}-s^{\left(r,c_{m}\right)}$,
$f^{\left(l\right)}=\{\begin{array}{ccc} \;\;\;\;\;\;\;\;\;\;\;-g^{\left(l\right)}-s^{\left(r,l\right)} & \mathrm{if} & \mathrm{lumped}\;\;\mathrm{mass},\\ -e^{\left(l\right)}-g^{\left(l\right)}-s^{\left(r,l\right)} & \mathrm{if} & \mathrm{sphere}\;\mathrm{or}\;\mathrm{box}, \end{array}$
$e^{\left(c\right)},\;\;e^{\left(l\right)}$ are the vectors of the Coriolis, gyroscopic, and centrifugal forces,
$g^{\left(c\right)},\;\;g^{\left(l\right)}$ are the vectors of the gravity forces,
$s^{\left(s\right)}$ is the vector of the spring and damping forces formulated to the wheels and outriggers,
$s^{\left(r,c_{m}\right)},\;\;s^{\left(r,l\right)}$ are the vectors of the spring and damping force(s) formulated to the rope(s),and 
$d^{\left(j\right)},\;\;d^{\left(d\right)}$ are the vectors of the right side of the constraints.

The computational procedure applied to derive the dynamics equations (Equation (2)) are partially presented in [25].

The statics analysis was performed based on the dynamics equations (Equation (2)), assuming the vectors of generalized accelerations (
${\ddot{q}}^{\left(c\right)},\;\;{\ddot{q}}^{\left(l\right)}$) and velocities (
${\dot{q}}^{\left(c\right)},\;\;{\dot{q}}^{\left(l\right)}$) were equal to zero and presenting the constraints equations in displacement forms. Then, a system of nonlinear algebraic equations was obtained. After solving them, the vectors of the generalized coordinates (
$q^{\left(c\right)},\;\;q^{\left(l\right)}$), the vector of the reaction forces at the cut-joints (
$f^{\left(j\right)}$), and the vector of the driving forces and torques (
$f^{\left(d\right)}$) are found.

## 3. Numerical Results

The statics equations were solved using the iterative Newton–Raphson method, in order to determine the initial configuration resulting from the flexibility of the system.

The initial conditions to the statics analysis are assumed as follows:(3)
$${q|}_{t=0}={\left[\begin{array}{ccccc} \underset{q^{{\left(b\right)}^{T}}}{\underset{︸}{0}} & \underset{q^{{\left(c_{m}\right)}^{T}}}{\underset{︸}{\begin{array}{cccc} -90{}^{\circ} & 0 & 270{}^{\circ} & 1.42\;m \end{array}}} & \underset{q^{{\left(c_{a,1}\right)}^{T}}}{\underset{︸}{\begin{array}{cc} 354.44{}^{\circ} & 0.38\;m \end{array}}} & \underset{q^{{\left(c_{a,2}\right)}^{T}}}{\underset{︸}{\begin{array}{cc} 92.16{}^{\circ} & 1.12\;m \end{array}}} & {q^{{\left(l\right)}^{T}}|}_{t=0} \end{array}\right]}^{T},$$
where

$${q^{\left(l\right)}|}_{t=0}=\{\begin{array}{ccc} \;\;\;\;\;\;\;\;\;\;\;{\left[\begin{array}{ccc} 0 & -5.80\;m & 0.75\;m \end{array}\right]}^{T} & \mathrm{if} & \mathrm{lumped}\;\;\mathrm{mass},\\ \;{\left[\begin{array}{cccccc} 0 & -5.80\;m & 0.75\;m & -90{}^{\circ} & 0 & 0 \end{array}\right]}^{T} & \mathrm{if} & \mathrm{sphere}\;\mathrm{or}\;\mathrm{box}. \end{array}$$

The fourth order Runge–Kutta method with a constant step size equal to 
$10^{-3}\;s$ for the system with a lumped mass and 
$10^{-4}\;s$ for the system with a sphere or a box was used to integrate the dynamics equations of motion.

The crane input was divided into five phases (Figure 3):
load lifting—movement performed by drive 
$d_{2}$ for 
$t\in \langle t_{iv}^{\left(1\right)},t_{fv}^{\left(1\right)}\rangle$;load telescoping—movement performed by drive 
$d_{4}$ for 
$t\in \langle t_{iv}^{\left(2\right)},t_{fv}^{\left(2\right)}\rangle$;load lowering—movement performed by drive 
$d_{3}$ for 
$t\in \langle t_{iv}^{\left(3\right)},t_{fv}^{\left(3\right)}\rangle$;crane rotation—movement performed by drive 
$d_{1}$ for 
$t\in \langle t_{iv}^{\left(4\right)},t_{fv}^{\left(4\right)}\rangle$;load swinging—for 
$t\in \langle t_{iv}^{\left(5\right)},t_{fv}^{\left(5\right)}\rangle$.

The initial and final values of the input motion are given as follows
(4a)
$$q_{a,iv}^{\left(d\right)}={\left[\begin{array}{cccc} \underset{q_{a,iv}^{\left(d_{2}\right)}}{\underset{︸}{0.38\;m}} & \underset{q_{a,iv}^{\left(d_{4}\right)}}{\underset{︸}{1.42\;m}} & \underset{q_{a,iv}^{\left(d_{3}\right)}}{\underset{︸}{1.12\;m}} & \underset{q_{a,iv}^{\left(d_{1}\right)}}{\underset{︸}{-90{}^{\circ}}} \end{array}\right]}^{T},$$
(4b)
$$q_{a,fv}^{\left(d\right)}={\left[\begin{array}{cccc} \underset{q_{a,fv}^{\left(d_{2}\right)}}{\underset{︸}{0.68\;m}} & \underset{q_{a,fv}^{\left(d_{4}\right)}}{\underset{︸}{0.92\;m}} & \underset{q_{a,fv}^{\left(d_{3}\right)}}{\underset{︸}{0.42\;m}} & \underset{q_{a,fv}^{\left(d_{1}\right)}}{\underset{︸}{0}} \end{array}\right]}^{T}.$$

The kinetic energy of the load is defined by the following formula
(5)
$$E_{k}^{\left(l\right)}=\{\begin{array}{ccc} \frac{1}{2}m^{\left(l\right)}\left({\dot{x}}^{{\left(l_{m}\right)}^{2}}+{\dot{y}}^{{\left(l_{m}\right)}^{2}}+{\dot{z}}^{{\left(l_{m}\right)}^{2}}\right) & \mathrm{if} & \mathrm{lumped}\;\;\mathrm{mass},\\ \frac{1}{2}\mathrm{tr}{\left({\dot{T}}^{\left(\alpha \right)}{\tilde{H}}^{\left(\alpha \right)}{\dot{T}}^{{\left(\alpha \right)}^{T}}\right)}_{\alpha \in \left\{l_{s},l_{b}\right\}} & \mathrm{if} & \mathrm{sphere}\;\mathrm{or}\;\mathrm{box}, \end{array}$$
where
$m^{\left(l\right)}$ is the mass of the load,
$T^{\left(\alpha \right)}$ is the transformation matrix describing the position and orientation of the load,
${\tilde{H}}^{\left(\alpha \right)}$ is the pseudo-inertia matrix of the load [26],and 
tr(⋅) is a trace of the matrix.

The time courses of kinetic energy of the three variants of the carried loads are shown in Figure 4.

The influence of load modeling methods on the kinetic energy of the load was estimated using following energy indicators:−kinetic energy integral mean value (Figure 5)
(6a)
$${{\bar{E}}_{k}^{\left(l\right)}|}_{t\in \left[t_{iv}^{\left(\alpha \right)},t_{fv}^{\left(\alpha \right)}\right]}=\frac{1}{t_{fv}^{\left(\alpha \right)}-t_{iv}^{\left(\alpha \right)}}\int_{t_{iv}^{\left(\alpha \right)}}^{t_{fv}^{\left(\alpha \right)}}E_{k}^{\left(l\right)}(t)dt$$−kinetic energy arithmetic average deviation from the integral mean value (Figure 6)
(6b)
$${{\hat{E}}_{k}^{\left(l\right)}|}_{t\in \left[t_{iv}^{\left(\alpha \right)},t_{fv}^{\left(\alpha \right)}\right]}=\frac{1}{n}\sum_{j=1}^{n}\left|E_{k}^{\left(l\right)}(t_{j})-{\bar{E}}_{k}^{\left(l\right)}\right|$$

Figure 7 and Figure 8 present the values of the indicators defined by Equation (6).

The relative percentage deviations of the maximum kinetic energy 
$E_{k,\max}^{\left(l\right)}$ and energy indicators 
${\bar{E}}_{k}^{\left(l\right)}$, 
${\hat{E}}_{k}^{\left(l\right)}$ from the values of these parameters obtained for the reference model, in which the load was treated as a lumped mass, can be calculated in the following form
(7a)
$${\delta E_{k,\max}^{\left(l\right)}|}_{l\in \{l_{s},l_{b}\},t\in \left[t_{iv}^{\left(\alpha \right)},t_{fv}^{\left(\alpha \right)}\right]}=\frac{E_{k,\max}^{\left(l\right)}-E_{k,\max}^{\left(l_{m}\right)}}{E_{k,\max}^{\left(l\right)}}\cdot 100\%,$$
(7b)
$${\delta {\bar{E}}_{k}^{\left(l\right)}|}_{l\in \{l_{s},l_{b}\},t\in \left[t_{iv}^{\left(\alpha \right)},t_{fv}^{\left(\alpha \right)}\right]}=\frac{{\bar{E}}_{k}^{\left(l\right)}-{\bar{E}}_{k}^{\left(l_{m}\right)}}{{\bar{E}}_{k}^{\left(l\right)}}\cdot 100\%,$$
(7c)
$${\delta {\hat{E}}_{k}^{\left(l\right)}|}_{l\in \{l_{s},l_{b}\},t\in \left[t_{iv}^{\left(\alpha \right)},t_{fv}^{\left(\alpha \right)}\right]}=\frac{{\hat{E}}_{k}^{\left(l\right)}-{\hat{E}}_{k}^{\left(l_{m}\right)}}{{\hat{E}}_{k}^{\left(l\right)}}\cdot 100\%.$$

The values of the relative percentage deviations of the kinetic energy and their indicators (Equation (7)) are presented in Table 2, Table 3 and Table 4.

## 4. Discussion

Based on the results presented in Figure 4 and Table 2, Table 3 and Table 4, the following conclusions can be formulated:The load modeling methods influence not only the maximum values but also the characters of the time courses of the kinetic energy in particular movement phases.The largest percentages of deviations in relation to the load reference model (load modeled as a lumped mass) are:−22.3% in phase-5, when the load was modeled in the form of a sphere;−23.0% in phase-3, when the load was modeled in the form of a box.

In the case of the load modeled in the form of a sphere, the value of kinetic energy was lower by 10% in phase-4.

The increase in the value of the mass of the load did not affect the percentage deviations.

The largest values of the relative percentage deviations of the kinetic energy integral mean values are:−22.5% in phase-5, when the load was modeled in the form of a sphere;−24.3% in phase-2, when the load was modeled in the form of a box.The largest values of the relative percentage deviations of the kinetic energy arithmetic average deviation from the integral mean values are: −23% in phase-5, when the load was modeled in the form of a sphere;−18.9% in phase-3, when the load was modeled in the form of a box.


The values of the kinetic energy-based indicators confirm that the method of modeling a load has significant influence on the dynamics of mobile cranes. In the models of mobile cranes used for analysis, e.g., accuracy of the positioning of the load after the lifting cycle, application of the load modeled as a rigid body (six dofs) is recommended. The increase in the value of the mass of the load does not affect the percentage difference.

In summary, the main contribution of this work is the indication that the use of a mathematical model of a crane with a load in the form of a lumped mass may not reflect the real dynamics of a crane, as proven by the values of the energy indicators shown in the numerical simulation. It should also be taken into account that the value of the kinetic energy of the load is often assumed to be one of the criteria in the optimization task.

## Figures and Tables

**Figure 1 materials-15-08156-f001:**
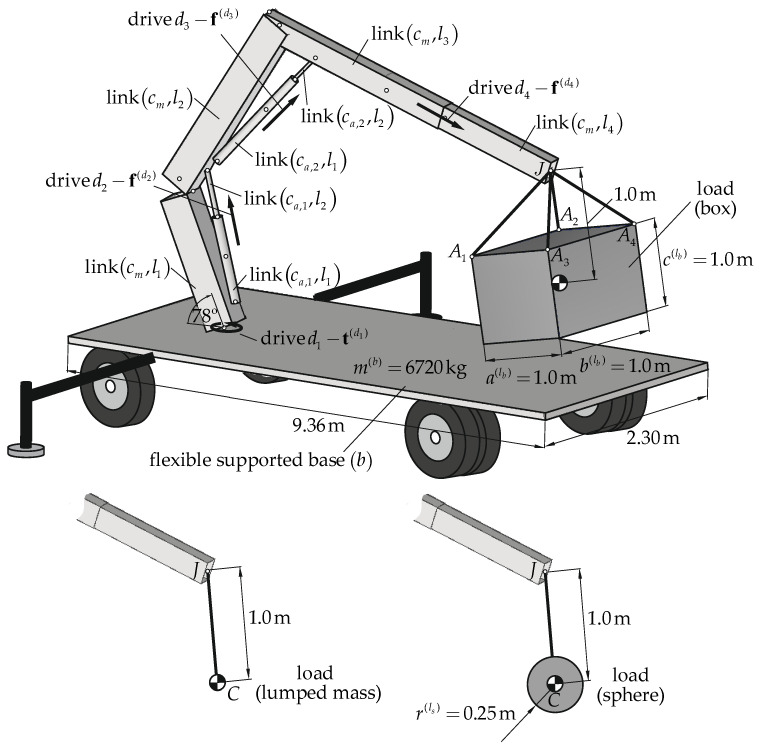
Model of the crane.

**Figure 2 materials-15-08156-f002:**
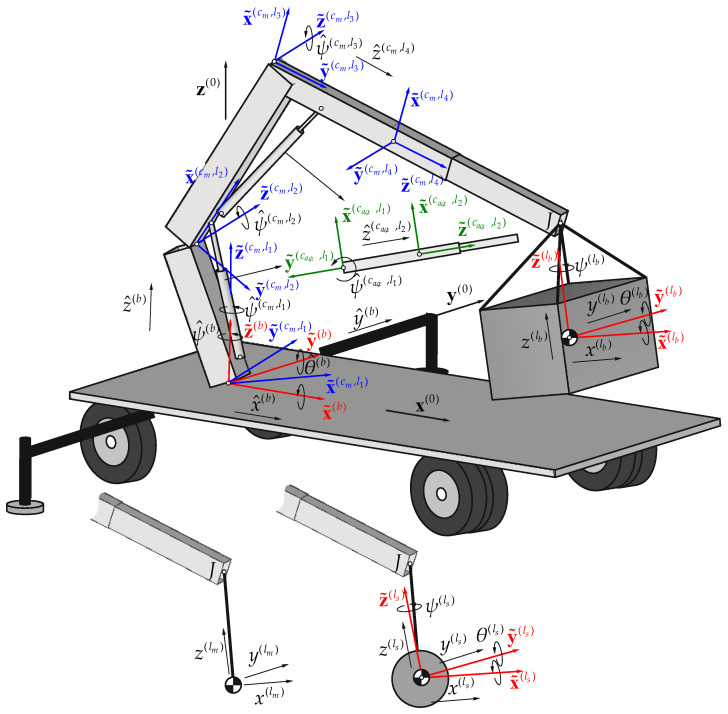
Description of the system—local frames of the links. Blue—main chain, green—auxiliary chains, red—flexible supported base and load.

**Figure 3 materials-15-08156-f003:**
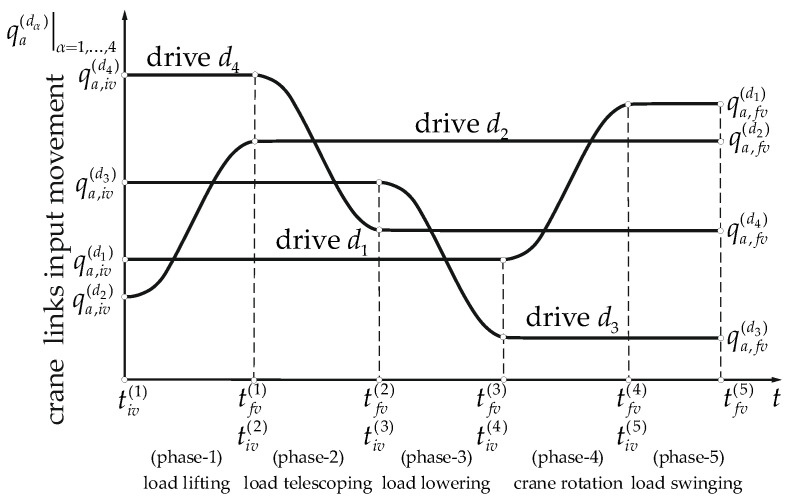
Crane input phases.

**Figure 4 materials-15-08156-f004:**
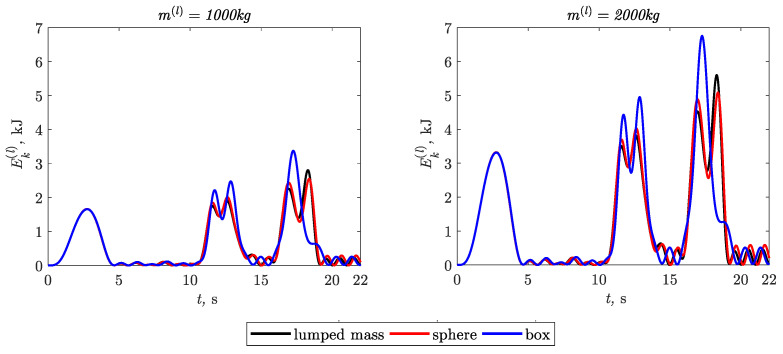
Time courses of the kinetic energy of the loads.

**Figure 5 materials-15-08156-f005:**
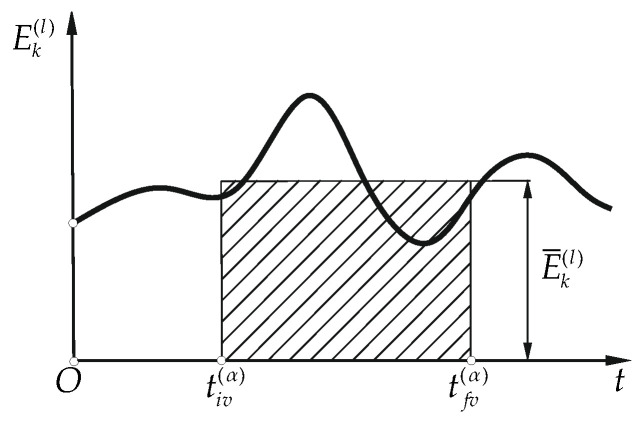
Kinetic energy integral mean value indicator.

**Figure 6 materials-15-08156-f006:**
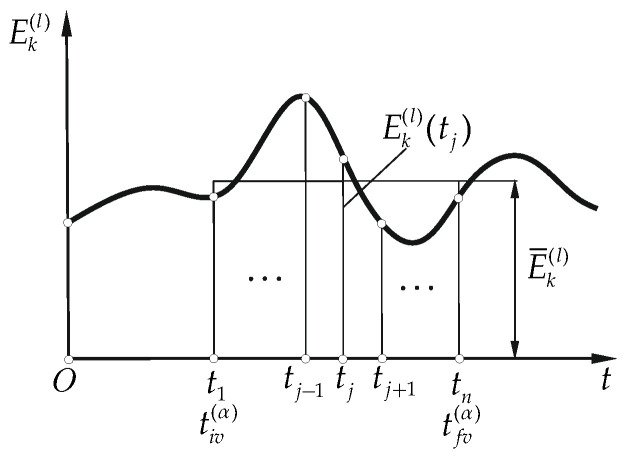
Kinetic energy arithmetic average deviation from the integral mean value indicator.

**Figure 7 materials-15-08156-f007:**
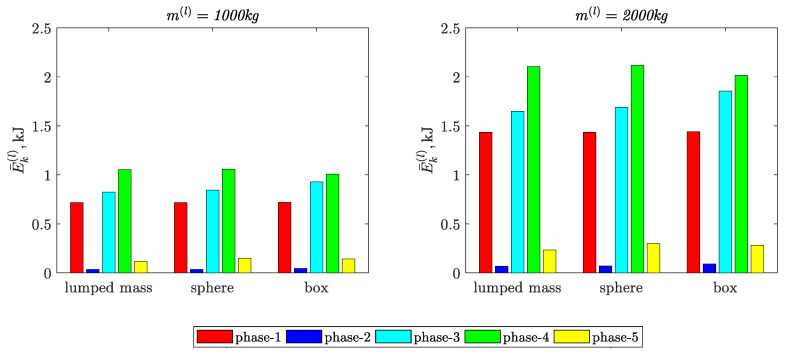
Kinetic energy integral mean value of the load in phases of input movement.

**Figure 8 materials-15-08156-f008:**
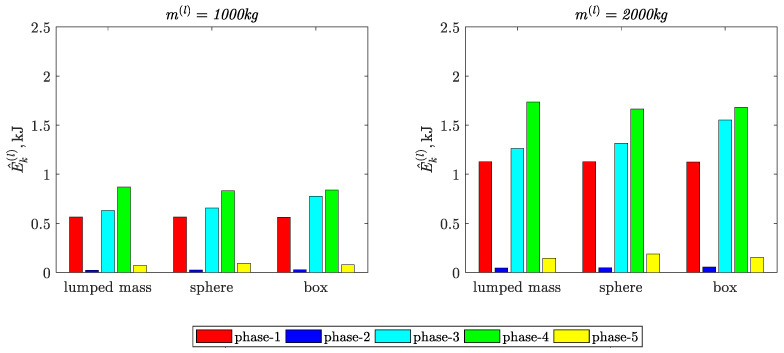
Kinetic energy arithmetic average deviation from the integral mean value of the load in the phases of input movement.

**Table 1 materials-15-08156-t001:** Parameters of the links.

Link	Length, m	Cross-Section, mm
$\left(c_{m},l_{1}\right)$	1.56	RHS 360×300×10
$\left(c_{m},l_{2}\right)$	2.45	RHS 360×280×6
$\left(c_{m},l_{3}\right)$	2.42	RHS 420×280×6
$\left(c_{m},l_{4}\right)$	2.25	RHS 270×170×5
$\left(c_{a,1},l_{1}\right)$	1.0	CHS 120×10
$\left(c_{a,1},l_{2}\right)$	1.0	CS 60
$\left(c_{a,2},l_{1}\right)$	1.5	CHS 100×10
$\left(c_{a,2},l_{2}\right)$	1.0	CS 40

**Table 2 materials-15-08156-t002:** Values of the kinetic energy in the movement phases and their relative percentage deviations.

		${E_{k,\max}^{\left(l\right)}|}_{t\in \left[t_{iv}^{\left(\alpha \right)},t_{fv}^{\left(\alpha \right)}\right]},\mathrm{kJ}\;\;;\;$ $\left({\delta E_{k,\max}^{\left(l\right)}|}_{t\in \left[t_{iv}^{\left(\alpha \right)},t_{fv}^{\left(\alpha \right)}\right]},\%\right)$				
		Phase-1	Phase-2	Phase-3	Phase-4	Phase-5
1000 kg	lumped mass	1.661	0.106	1.907	2.805	0.227
	sphere	1.660; (−0.04)	0.108; (1.25)	2.000; (5.13)	2.550; (−10.01)	0.292; (22.32)
	box	1.656; (−0.31)	0.117; (8.93)	2.474; (22.93)	3.373; (16.85)	0.258; (11.89)
2000 kg	lumped mass	3.323	0.213	3.819	5.599	0.458
	sphere	3.322; (−0.04)	0.216; (1.27)	4.027; (5.18)	5.084; (−10.14)	0.590; (22.28)
	box	3.312; (−0.33)	0.234; (8.89)	4.950; (22.85)	6.752; (17.07)	0.513; (10.61)

**Table 3 materials-15-08156-t003:** The kinetic energy integral mean values in the movement phases and their relative percentage deviations.

		${{\bar{E}}_{k}^{\left(l\right)}|}_{t\in \left[t_{iv}^{\left(\alpha \right)},t_{fv}^{\left(\alpha \right)}\right]},\mathrm{kJ}\;;\;$ $\left({\delta {\bar{E}}_{k}^{\left(l\right)}|}_{t\in \left[t_{iv}^{\left(\alpha \right)},t_{fv}^{\left(\alpha \right)}\right]},\%\right)$				
		Phase-1	Phase-2	Phase-3	Phase-4	Phase-5
1000 kg	lumped mass	0.717	0.034	0.824	1.052	0.115
	sphere	0.718; (0.06)	0.035; (3.45)	0.845; (2.41)	1.058; (0.62)	0.149; (22.50)
	box	0.720; (0.43)	0.045; (24.38)	0.927; (11.09)	1.008; (−4.35)	0.142; (18.46)
2000 kg	lumped mass	1.434	0.068	1.650	2.105	0.233
	sphere	1.435; (0.06)	0.071; (3.46)	1.691; (2.41)	2.117; (0.59)	0.300; (22.39)
	box	1.440; (0.42)	0.090; (24.37)	1.855; (11.04)	2.016; (−4.38)	0.282; (17.34)

**Table 4 materials-15-08156-t004:** The kinetic energy arithmetic average deviation from the integral mean values in the movement phases and their relative percentage deviations.

		${{\hat{E}}_{k}^{\left(l\right)}|}_{t\in \left[t_{vi}^{\left(\alpha \right)},t_{fv}^{\left(\alpha \right)}\right]},\mathrm{kJ}\;\;;$ $\left({\delta {\hat{E}}_{k}^{\left(l\right)}|}_{t\in \left[t_{iv}^{\left(\alpha \right)},t_{fv}^{\left(\alpha \right)}\right]},\%\right)$				
		Phase-1	Phase-2	Phase-3	Phase-4	Phase-5
1000 kg	lumped mass	0.563	0.023	0.630	0.869	0.071
	sphere	0.563; (0)	0.024; (3.62)	0.657; (4.18)	0.833; (−4.33)	0.092; (23.04)
	box	0.563; (−0.10)	0.028; (17.41)	0.776; (18.87)	0.839; (−3.51)	0.078; (8.54)
2000 kg	lumped mass	1.127	0.046	1.261	1.736	0.144
	sphere	1.127; (−0.01)	0.048; (3.62)	1.316; (4.18)	1.664; (−4.35)	0.187; (23.00)
	box	1.126; (−0.10)	0.056; (17.35)	1.553; (18.79)	1.682; (−3.21)	0.155; (7.08)

## Data Availability

The data presented in this study are available upon request from the corresponding authors.

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
