# Peer review of "Kinetic Energy-Based Indicators to Compare Different Load Models of a Mobile Crane"

_materials, 2022, doi:10.3390/ma15228156_

Round 1

Reviewer 1 Report

The article presents a proposal for using energy indicators to evaluate the load modelling methods on the dynamics of a mobile crane. The article does not really carry any essential contribution. It is a simple application and there is not a much-presented contribution for interested readers. The research is more or less just a mathematical model example of an optimization problem. The data used here are very limited and do not give an essential contribution to the basic knowledge of the material. The technical writing of this manuscript is poor. The writings do not effectively describe nor explain the key results of this study. The study lacks a literature review which is important for the reader to understand the topic. The Introduction section includes 3 short paragraphs with insufficient details in each sub-topic, the introduction lacks the problem statement of this research, motivations, research gap, novelty, and implications of this study. The methodology of this study is not clear; you may use a flow chart to describe the main steps you used to perform this analysis.  The limitations and scope of this study should be addressed. For the section “Conclusions”, you should explain the problem statement, merits, motivations, and approach of this study.   The references should be from the web of science 2020-2022 (50% of all references). The implications and feasibility of this work should also be discussed. The article presents a proposal for the use of energy indicators to evaluate the load modeling methods on the dynamics of a mobile crane. The article does not really carry any essential contribution. It is a simple application and there is not a much-presented contribution for the interested readers. The research is more or less just mathematical models example of an optimization problem. The data used here are very limited and do not give an essential contribution to the basic knowledge of material. The technical writing of this manuscript is poor. The writings do not effectively describe nor explain the key results of this study. The study lacks a literature review that is important for the reader to understand the topic. The Introduction section includes 3 short paragraphs with insufficient details in each sub-topic, the introduction lacks the problem statement of this research, motivations, research gap, novelty, and implications of this study. The methodology of this study is not clear; you may use a flow chart to describe the main steps you used to perform this analysis.  The limitations and scope of this study should be addressed. For the section “Conclusions”, you should explain the problem statement, merits, motivations, and approach of this study.   The references should be from the web of science 2020-2022 (50% of all references). The implications and feasibility of this work should also be discussed. My final opinion is to reject the article.

Author Response

The answers are in the attached file.

Reviewer 2 Report

The paper deals with a proposal for the use of energy indicators to evaluate the load modeling methods on the dynamics of a mobile crane. Three different variants of mathematical models of a load carried are examined and compared. The equations of motion are derived using the Lagrange equations of the second kind. These equations are supplemented by the Lagrange multipliers and constraint equations formulated for the cut-joints and drives.

The paper contributes to the knowledge base. The paper is written in an appropriate way.

I do not suspect the authors of plagiarism, fraud or other ethical concerns. The English language is appropriate and understandable.

The language is fluent, but in my opinion some sequential steps of deriving the equations could be commented in more detail.

Specific comments on the paper are given below.

Comments:

1. Introduction: The state-of-the-art should be presented in more detail. This refers to summary citations [5-10] and [16-19].

2. Row 40: Describe specifically the mentioned four main structures in Figure 1.

3. Figure 1: It is not evident what abbreviations RHS, CHS and CS mean. Describe meaning of parameters and variables used in Figure 1 in the paper.

4. Row 101: Explain verbally in the text of the paper the five phases of the crane input movement.

5. Row 105: Formula for calculating kinetic energy is not presented in the paper.

6. Row 106: The time courses of kinetic energy are presented in Figure 4, not in Figure 3.

Formal comments:

1. References: The format of some references is not in compliance with the requirements of the Materials Journal.

Author Response

The answers are in the attached file.

Reviewer 3 Report

Report on the Manuscript materials-2021938

Title: " Kinetic energy based indicators to compare different load models of a mobile crane"

 The aim of this study is to present a proposal for the use of energy indicators to evaluate the load modeling methods on the dynamics of a mobile crane. The following comments must be taken into consideration.

1.      A native English speaker would be useful to check the whole of the text for grammatical style and word use.

2.      Keywords must be revised a little more and extended.

3.      Too much data on the drawing, which increases the density of the figure and this, leads to misunderstanding.

4.      Poor resolution of the included figures is observed.

5.      Short notes about “a dynamical model for load-lifting system of the crane” must be included in the manuscript.

6.      What about the influence of various masses on the obtained results?.

7.      Authors should clear statements of the novelty of the work, should also appear briefly in the Abstract and Conclusions sections.

8.      Authors should improve the introduction by including the recent development within the frame of the corresponding analytical solutions of different motions of bodies with their stabilities by considering the help of recently published papers.

·         https://doi.org/10.1016/j.apm.2020.08.008

·         https://doi.org/10.1016/j.rinp.2019.01.037

9.      Have the authors employed any assumptions on model? Please explain briefly.

10.  What about the stability of the proposed model?.

11.  Concluding Remarks section must be extended so that it provides and covers all the finding of the paper and future direction. Moreover it should mention some important meaning of simulations as conclusion.

Author Response

The answers are in the attached file.

Reviewer 4 Report

In this paper, the authors presented a method that uses the energy indicator to evaluate the load modeling method on the dynamics of a mobile crane. The article is well-written and the method results are presented clearly. The article can be accepted in the current version with a minor revision. The following are my suggestions:

1. In the introduction section, the authors should discuss the application and limitations briefly,  which is to emphasize the importance of the current work.

2. The novelty of the paper (advantage of the method) should be stated in detail in the introduction or discussion section.

3. The numerical algorithm (pseudocode) to solve the static equations should be appended in the appendix. 

Author Response

The answers are in the attached file.

Round 2

Reviewer 1 Report

I have no more comments on this revision and could be accepted for publication. Accept in present form

Reviewer 2 Report

All my comments were accepted.

Reviewer 3 Report

This paper can be accepted in the the present form